# Removing False Targets for Cyclic Prefixed OFDM Sensing with Extended Ranging [note 1]

**DOI:** 10.3390/s22229015

**Published:** 2022-11-21

**Authors:** Kai Wu, J. Andrew Zhang, Xiaojing Huang, Y. Jay Guo

**Affiliations:** Global Big Data Technologies Centre (GBDTC), University of Technology Sydney (UTS), Sydney, NSW 2007, Australia

**Keywords:** integrated sensing and communications (ISAC), joint communications and sensing (JCAS), OFDM, cyclic prefix (CP), false targets

## Abstract

Employing a cyclic prefixed OFDM (CP-OFDM) communication waveform for sensing has attracted extensive attention in vehicular integrated sensing and communications (ISAC). A unified sensing framework was developed recently, enabling CP-OFDM sensing to surpass the conventional limits imposed by underlying communications. However, a false target issue still remains unsolved. In this paper, we investigate and solve this issue. Specifically, we unveil that false targets are caused by periodic cyclic prefixes (CPs) in CP-OFDM waveforms. We also derive the relation between the locations of false and true targets, and other features, e.g., strength, of false targets. Moreover, we develop an effective solution to remove false targets. Simulations are provided to confirm the validity of our analysis and the effectiveness of the proposed solution. In particular, our design can reduce the false alarm rate caused by false targets by over 50% compared with the prior art.

## 1. Introduction

Integrated sensing and communications (ISAC) has attracted extensive attention in the radar and communications community [1,2,3,4,5,6,7,8]. A key benefit of ISAC is that it can greatly improve cost, energy, and spectral efficiency of wireless systems that require both sensing and communications functions [9]. While there are several ways of realizing ISAC, as reviewed in [10,11], a relatively easy way is to add a sensing receiver to an existing communication system and perform sensing by reusing the communication waveforms. We refer to this ISAC methodology as the communications-enabled sensing (CES) and focus on it in this work. Interested readers are referred to [12] for a review of CES.

Orthogonal frequency-division multiplexing (OFDM) and its variants, e.g., DFT-spread OFDM (DFT-S-OFDM), have been widely studied for CES. In [13], OFDM CES is considered, with a focus on evaluating the impact of radar antenna setups on common communication metrics. Before the work of [13], OFDM radar was extensively studied without considering data communications in general [14]. The ambiguity functions of OFDM radars were intensively studied as well [15,16,17,18,19,20].

A milestone CES method was developed in [21], referred to as the classical OFDM sensing (COS). Using a coherent synchronised transceiver, COS removes the cyclic prefix (CP) of each OFDM symbol in the echo signal. Then, it transforms the symbols into the frequency domain, where the communication data symbols modulated onto sub-carriers are removed through point-wise division (PWD). Afterwards, COS performs inverse discrete Fourier transforms (IDFTs) along the sub-carrier dimension and discrete Fourier transforms (DFTs) over the symbol dimension, resulting in a range-Doppler map (RDM).

COS has been widely used in the past decade and has become a mainstream for OFDM sensing, particularly in automotive applications [7,8,22,23,24,25,26,27,28,29,30]. Recent OFDM sensing works, which are based on COS, mainly seek to develop new techniques for improving the RDM quality, such as noise level. The work in [23] increased the overall baseband bandwidth of the OFDM by introducing the stepped carrier techniques. This, hence, improves the range resolution. The work in [24] employed random stepped carriers and the compressed sensing methods to obtain high-resolution RDMs. The works reviewed above typically ignore the intercarrier interference (ICI). In contrast, the work in [25] considered the impact of ICI on OFDM sensing and developed a novel signalling, which repeats the same OFDM symbol over (slow-)time, to facilitate the estimation and suppression of ICI.

The phase shift keying was considered in [21], and hence PWD does not have a noise enhancement issue. However, this issue can be prominent when DFT-S-OFDM is performed. This is because signals modulated onto sub-carriers in DFT-S-OFDM can be noise-like and centred around zero. To relieve noise enhancement, the work in [31] proposed to replace PWD with the point-wise product (PWP) and followed the other steps of COS. While COS and PWP-COS fully comply with the underlying communication system, their ranging capability is known to be restricted by the CP duration.

Such restriction is relieved by a unified sensing framework (USF) developed in [32] (the unification is based on the feature that the sensing framework can be applied to OFDM, DFT-S-OFDM, and other OFDM-like waveforms, e.g., the orthogonal time-frequency space (OTFS), even with reduced CP (RCP). RCP-OTFS adds a single CP for the signal block that may be equivalent to multiple OFDM symbols. COS and many of its variants cannot work for RCP waveforms). USF segments a block of sensing echo signal as per sensing needs and employs virtual CPs to establish a deterministic relation between the transmitted signal and each segmentation of echo signals. A notable feature of USF is that it can sense a target with the sample delay much greater than CP length (provided there is a sufficient link budget). More details of USF will be presented in Section 2.

However, as illustrated in [12,32], USF suffers from a false target issue. Specifically, obvious peaks (much stronger than background noises) are observed at locations different from that of the true target. In a previous work [33], we intuitively explained the reasons for false targets employing some special cases, and developed a simple expedient solution that can cause a decrease of use signal power and, hence, the detecting probability. In this work, we perform an in-depth investigation to more rigorously unveil the reasons for the false target issue in the USF. Moreover, we will develop more effective solutions to removing false targets without losing the power of useful signals.

Our main contributions are summarised below. We introduce a concise matrix format of USF, which conveniently enables the investigation into the false target issue. We unveil that false targets are mainly caused by the periodic CP signals in the CP-OFDM waveform. We also derive the properties of false targets, including positions and strengths. Accordingly, we develop novel methods, removing the false targets without decreasing the detecting probability of true targets. Simulations are provided to confirm the validity of our analysis and the effectiveness of the proposed solution.

The rest of the paper is arranged as follows. Section 2 establishes the signal model. Section 3 first reviews USF and another classical OFDM sensing method and then illustrates the false target problem. Section 4 investigates the reasons for and features of false targets. Section 5 develops two novel methods for handling false targets. Simulation results are presented in Section 6, and conclusions are given in Section 7.

## 2. Signal Model

This section presents the signal model of OFDM-ISAC. Consider that a communication node is turned into an ISAC node by adding a sensing receiver. Sensing is performed using the echo signals resulted from the communication-transmitted signals. Being co-located, the communication transmitter and sensing receiver are assumed to be fully synchronised. It is also reasonable to assume that the sensing receiver has a copy to the time-domain communication-transmitted signals. It is further assumed that some proper full-duplex techniques are applied such that self-interference, i.e., the signal leaking from the transmitter to receiver, can be ignored. As for waveform, we employ OFDM with regular CP (i.e., one per symbol), which is widely used in modern communication systems.

We consider *M* consecutive OFDM symbols, each having *N* sub-carriers. Let *Q* denote the CP length. After CP is added, the time-domain signal matrix can be given by [34]
(1)$$\begin{array}{cc} \;\tilde{X}=\left[\begin{array}{c} V\\ X \end{array}\right],\;s.t.\;X={\left[\begin{array}{cccc} u_{0} & u_{1} & \cdots & u_{M-1}\\ \;v_{0} & v_{1} & \cdots & v_{M-1} \end{array}\right]}_{N\times M}\\ \; & \;\;\;\;\;\;\;\;\;\;\;\;\;\;\;\;\;\;\;\;V={\left[v_{0}\;v_{1}\;\cdots \;v_{M-1}\right]}_{Q\times M} \end{array}$$
where each column of $\tilde{X}$ represents an OFDM symbol, $v_{m}\;\left(\forall m\right)$ denotes CP, and $\left[\begin{array}{c} u_{m}\\ v_{m} \end{array}\right]\;\left(\forall m\right)$ is often referred to as the essential signal part of each OFDM symbol.

Reshaping $\tilde{X}$ into a vector by stacking its columns, we obtain
(2)$$\begin{array}{c} x=\mathrm{vec}\left\{\tilde{X}\right\}. \end{array}$$
At the communication transmitter, x is transmitted after being processed by a radio frequency (RF) chain. The RF signal will hit some targets and partially return to the sensing receiver. After some RF processing, the digital signal at the receiver can be given by [11]
(3)$$\begin{array}{c} y=\alpha S\mathrm{vec}\left\{\tilde{X}\right\}⊙d+z,\;s.t.\;S=\left[\begin{array}{c} 0_{L\times I}\\ I_{(I-L)\times I} \end{array}\right],\;{\left[d\right]}_{i}=e^{j2\pi \mu iT_{s}}, \end{array}$$
where α denotes the scattering coefficient, ⊙ denotes the point-wise product, and z denotes a noise vector with independent entries conforming to the same centred complex Gaussian distribution. Moreover, *L* denotes the sample delay, I=(N+Q)M is the total number of samples in a block, $I_{(I-L)\times I}$ denotes the first (I−L) rows of the *I*-dimensional identity matrix, d accounts for the Doppler impact, ${\left[\cdot \right]}_{i}$ takes the *i*-th entry of the enclosed vector, μ denotes the Doppler frequency, and $T_{s}$ the sampling time. A single target is modelled here for clarity. The multitarget case will be considered in later algorithm development and simulation.

We emphasise that, in the considered ISAC, sensing is performed based on the echo signal at the receiver, i.e., y, and the known copy of the communication-transmitted signals, i.e., $\tilde{X}$. That is, the ISAC scheme discussed here does not make any changes to the communication waveform. Moreover, we perform ISAC based on the predominant OFDM communication system that is widely under use in WiFi, LTE, and 5G. ISAC employing more advancing OFDM systems, such as the real-signal DHT-OFDM [35], NOMA-TDS-OFDM [36,37], and the precoded IM-OFDM-SS [38], can be an interesting future work.

## 3. Problem Statement

In this section, we briefly review USF [32] and then illustrate the false target issue. Unlike in the original work using the scalar form for developing the algorithm, we introduce a more concise matrix form here. The new matrix form will also benefit the analysis to be performed in Section 4. To start with, we define the following operator, reshaping a vector into a matrix.

**Definition** **1.**
*The operator $\underset{x,y}{\mathrm{mat}}\left\{c\right\}$ reshapes the vector c into a matrix of size $x\times \left\lfloor \frac{C-y}{x-y}\right\rfloor$, where each column has x number of entries and starts from the (i(x−y)+1)-th $\left(i=0,1,\cdots ,\left\lfloor \frac{C-y}{x-y}\right\rfloor -1\right)$ entry in c. Here, $\left\lfloor \cdot \right\rfloor$ takes flooring, C denotes the dimension of c, x is a positive integer (PI), and y can be a PI or zero.*


Note that, if *y* is nonzero, adjacent columns of the resulted matrix have overlapping entries. To help understand the operator, let us look at an example. Given $c={\left[1,2,3,\cdots ,9\right]}^{T}$, $\underset{4,1}{\mathrm{mat}}\left\{c\right\}$ and $\underset{4,2}{\mathrm{mat}}\left\{c\right\}$, respectively, result in
$$\begin{array}{c} \left[\begin{array}{cc} 1 & 4\\ 2 & 5\\ 3 & 6\\ 4 & 7 \end{array}\right]\;\;\mathrm{and}\;\;\left[\begin{array}{ccc} 1 & 3 & 5\\ 2 & 4 & 6\\ 3 & 5 & 7\\ 4 & 6 & 8 \end{array}\right] \end{array}$$

Based on the signal model given in Section 2, USF [32] is summarised in Algorithm 1. As performed in Step 1, the echo signal vector y is reshaped into a matrix by $\underset{\tilde{N}+\tilde{Q},\bar{Q}+\tilde{Q}}{\mathrm{mat}}\left\{y\right\}$, where each symbol has $\left(\tilde{N}+\tilde{Q}\right)$ samples and adjacent symbols are overlapped by $\left(\bar{Q}+\tilde{Q}\right)$ samples. Note that $\left(\tilde{N}+\tilde{Q}\right)$ here can be different from (N+Q), the length of the original OFDM symbol (with CP). In Step 2, the communication-transmitted signal, x given in (Equation 2), is reshaped into a matrix by $\underset{\tilde{N},\bar{Q}}{\mathrm{mat}}\left\{x\right\}$. We refer interested readers to Figure 5 in [12] for an intuitive illustration of the reshaping performed above. As a result of Steps 1 and 2, each column of $\tilde{Z}$, as obtained in Step 3, contains a cyclically shifted version of the corresponding column in $\overset{˘}{X}$, as obtained in Step 2. The shifting amount is linked with the target delay. Therefore, each column of $\bar{Z}$, as obtained in Step 4, is the point-wise product (PWP) between the same column of $\bar{X}$, as obtained in Step 5, and the range steering vector of the target.
**Algorithm** **1** Unified sensing framework (USF) [32].1.        Reshape y by $\underset{\tilde{N}+\tilde{Q},\bar{Q}+\tilde{Q}}{\mathrm{mat}}\left\{y\right\}$, leading to Z;

2.        Reshape x by $\underset{\tilde{N},\bar{Q}}{\mathrm{mat}}\left\{x\right\}$, leading to $\overset{˘}{X}$;3.        Add the last $\tilde{Q}$ rows of Z onto its first ones and remove the last $\tilde{Q}$ rows, leading to $\tilde{Z}$;4.        Take the DFT of the columns of $\tilde{Z}$, giving $\bar{Z}$;5.        Take the DFT of the columns of $\overset{˘}{X}$, giving $\bar{X}$;6.        Calculate the point-wise product (PWP) between $\bar{Z}$ and ${\bar{X}}^{*}$, yielding $\hat{Z}$, where ${\left(\cdot \right)}^{*}$ denotes conjugate;7.        Take the IDFT of the columns of $\hat{Z}$ and then the DFT of the rows, yielding an RDM $\overset{˘}{Z}$.

A point-wise division (PWD) between $\bar{Z}$ and $\bar{X}$ can remove the communications information, making the extraction of sensing target parameters independent of communications information. However, when the entries of $\bar{X}$ approach zero, the division can lead to the noise enhancement issue. To relieve the issue, it was proposed in [31] to replace PWD with PWP. As proved in our previous work [32], PWP theoretically achieves higher signal-to-interference-plus-noise ratio (SINR) than PWD, given high to medium noise power; the threshold for determining the noise level is also derived therein. This makes PWP more desirable in practice. Hence, we mainly consider PWP here.

Step 7 of Algorithm 1 performs a two-dimensional Fourier transform of the PWP result, leading to the so-called range-Doppler map (RDM). Here, the discrete Fourier transform (DFT) along the symbol domain turns a Doppler steering vector into a discrete Doppler spectrum. The inverse DFT (IDFT) over sub-carriers turns an amplitude-weighted range steering vector into a discrete range spectrum. As a result of PWP in Step 6, the amplitude weights are the squared amplitudes of communications signals over sub-carriers.

If a constant-modulus constellation, such as phase shift keying (PSK), is used, the amplitude weights become all ones. The discrete range spectrum can be depicted using a discrete sinc function. In contrast, if signals carried by sub-carriers have nonconstant amplitudes, the amplitude weighting on the range steering vector can make the discrete range spectrum different from a sinc function in shape. However, the impact of the weighting is not deterministic, as signals on sub-carriers are generally random in actual communications. It is also this randomness that prevents the sidelobe level in the RDM being constructively accumulated. We refer interested readers to [32] for a more elaborate analysis on the impact of the above amplitude weighting.

The classical OFDM sensing (COS) can be seen as a special case of USF and is summarised in Algorithm 2. COS fully complies with the underlying data communications. The echo signal vector y is reshaped first in Step 1. However, different from Step 1 of Algorithm 1, the length of each segmentation is (N+Q), identical to the length of an OFDM symbol (with CP). Different from Step 3 of Algorithm 1 creating virtual CPs, Step 2 of Algorithm 2 removes the first *Q* samples per symbol, as performed in OFDM communication receivers. Note that X in Step 4 is the original frequency-domain communication signals. Thus, COS need not reshape communication-transmitted signals, as required in USF.
**Algorithm** **2** Classical OFDM sensing (COS) [21].1.        Reshape y by $\underset{N+Q,0}{\mathrm{mat}}\left\{y\right\}$, leading to Y;2.        Remove the first *Q* rows of Y, leading to $\tilde{Y}$;3.        Take the DFTs of the columns of $\tilde{Y}$, giving $\bar{Y}$;4.        Calculate the PWP between $\bar{Y}$ and $X^{*}$, yielding $\hat{Y}$;5.        Take the IDFT of the columns of $\hat{Y}$ and then the DFT of the rows, yielding an RDM $\overset{˘}{Y}$.

Without fully complying with the communications system, USF can achieve more flexible sensing than COS, through configuring $\tilde{N}$, $\tilde{Q}$, and $\bar{Q}$ in Algorithm 1. Note that $\tilde{Q}$ is the length of a virtual CP created by the sensing receiver and can be configured based on sensing needs. More specifically, if the maximum sample delay is *L*, we can set $\tilde{Q}\geq L$ to satisfy this requirement. In contrast, L≤Q is generally required in COS [21] and its variant [31], where *Q* is the original CP length for communications. We remark that the value of $\tilde{Q}$ can affect the signal-to-interference-plus-noise ratio (SINR) in the obtained RDM. This is detailedly analysed in [32]. Here, we shall mainly focus on the false target issue that has not been effectively solved.

Given the maximum measurable Doppler frequency requirement $\mu_{\max}$, we can set $\tilde{N}$ such that $\frac{1}{\tilde{N}T_{s}}\geq 2\mu_{\max}$. In contrast, COS and many of its variants have their maximum measurable Doppler frequency limited to $\frac{1}{2NT_{s}}$, and hence they will suffer from Doppler ambiguity if $\mu_{\max}>\frac{1}{2NT_{s}}$.

Next, we provide a set of simulation results to demonstrate the superiority of USF over COS in extended sensing capability, meanwhile illustrating the false target issue of USF. Simulation parameters are summarised in Table 1. The target delay is much larger than the CP length, which is purposely set to show the extended sensing capability of USF over COS.

Figure 1a,b illustrate the RDMs of USF and COS, as obtained by running Algorithms 1 and 2, respectively. We see that COS fails to generate a normal RDM for target detection and estimation, while USF yields a typical RDM with a peak at the true target location. Note that the two algorithms are performed employing the same communication signals and echo signals. We note that all steps in the two algorithms are solely performed at the sensing receiver side without making changes to the communication transmitter.

Figure 2 plots the range and Doppler cuts of the RDMs obtained in Figure 1. Given the sample delay L=N+Q, the range bin (in the sensing receiver, each OFDM symbol is uniformly sampled. Each of the sampling times represents a different range increment and is often referred to as a range bin [39]) index for USF is *L* (starting from one here to comply with MATLAB). For COS, since the overall number of samples along the range bin is *N*, smaller than *L*, a modulo-*N* of *L*, i.e., (L−N) here, is taken as the range bin index for COS. The Doppler bin (similar to the range bin, the DFT of OFDM symbols at the same range bin leads to the Doppler domain. Each discretisation grid in the Doppler domain is termed as a Doppler bin) index can be calculated as $\left\lfloor \mu T\right\rceil +1$, where $\left\lfloor \cdot \right\rceil$ rounds the enclosed term, “+1” is because the index starts from one, $T=M\left(N+Q\right)T_{s}$ for COS, and $T=\tilde{M}\tilde{N}T_{s}$. Here, $\tilde{M}$ is the number of columns of Z obtained in Step 1 of Algorithm 1. According to Definition 1, we have $\tilde{M}=\left\lceil \frac{(N+Q)M}{\tilde{N}}\right\rceil$.

From Figure 2, we see strong peaks in the range and Doppler cuts of the USF-RDM. In contrast, we see noise-like signals over the whole range and Doppler bins in the cuts of the COS-RDM. Given the high peak-to-sidelobe ratio at the true target location, the target can be readily detected employing common detecting algorithms, e.g., the constant false alarm rate (CFAR) detector [40]. However, from Figure 2, we also see two false targets around the true one along the range dimension. This problem was noticed in [32], yet is unsolved. Below, we first investigate the reasons for the false targets and then develop new methods to solve the problem.

## 4. Investigating Reasons for False Targets

We analyse the reasons for for the false targets in this section. In particular, we start with a special case to understand the issue intuitively and then generalise the observation with more strict analysis.

### 4.1. A Special Case for Looking into False Targets

From Figure 1a, we see that the false targets exist over the range dimension solely. This indicates that they are generated by some column operations in Algorithm 1. Let us consider a special case with L=N, $\tilde{Q}=N+Q$, $\tilde{N}=2\tilde{Q}$, $\bar{Q}=0$, α=1, and μ=0. From (Equation 3), we see that μ=0 leads to d=1. Based on these settings, the first column of $\overset{˘}{X}$, as obtained in Step 2 of Algorithm 1, can be given by
(4)$$\begin{array}{c} e=\left[v_{0};u_{0};v_{0};v_{1};u_{1};v_{1}\right], \end{array}$$
where “;” concatenate column vectors. As given in (Equation 1), the dimensions of the *u*-vector and *v*-vector are (N−Q) and *Q*, respectively. For ease of illustration, the noise term will be ignored in this section, as the false target issue exists regardless of noises.

Given L=N and the expression of y in (Equation 3), the first column of Z obtained in Step 1 of Algorithm 1 can be given by
$$\begin{array}{c} \left[0_{N\times 1};v_{0};u_{0};v_{0};v_{1};u_{1};v_{1};v_{2}\right], \end{array}$$
where $0_{N\times 1}$ denotes an N×1 zero vector and the zero Doppler is assumed. The above vector, going through Step 3 of Algorithm 1, becomes
(5)$$\begin{array}{cc} \; & f=\underset{{\langle e\rangle }_{N}}{\underset{︸}{\left[u_{1};v_{1};v_{0};u_{0};v_{0};v_{1}\right]}}+\left[0_{N\times 1};v_{2};0_{(N+Q)\times 1}\right], \end{array}$$
where ${\langle e\rangle }_{N}$ indicates that the vector e is circularly shifted by *N* entries. Note that the second vector on the right-hand side of (Equation 5) is an interference term. It is a price paid by USF to relieve the constraint L≤Q, as suffered by COS, given in Algorithm 2. The calculations in Steps 5–7 of Algorithm 1 make the first column of $\overset{˘}{Z}$ (the RDM) become the cyclic cross-correlation (CCC) between e and f. Let g denote the first column of $\overset{˘}{Z}$. Its *i*-th entry satisfies
(6)$$\begin{array}{c} {\left[g\right]}_{i}={\left({\langle e\rangle }_{i}\right)}^{H}f\approx {\left({\langle e\rangle }_{i}\right)}^{H}{\langle e\rangle }_{N}. \end{array}$$

**Remark** **1.**
*We remark on the features of $\left[\begin{array}{c} u_{m}\\ v_{m} \end{array}\right]\;\left(\forall m\right)$, as given in (Equation 1). In particular, the entries of $\left[\begin{array}{c} u_{m}\\ v_{m} \end{array}\right]$ approximately converge in distribution to a complex Gaussian random process and it is independent over m, provided that the communication symbols carried by sub-carriers are randomly and uniformly drawn from some constellations. The proof can be readily established based on Remark 2 in [41] and is hence suppressed here for brevity. Applying the above feature, we have that $v_{2}$ in f is independent from any consecutive Q entries in e. This leads to approximation in (Equation 6).*


From (Equation 4) and (Equation 5), we see that ${\left[g\right]}_{i}$ is maximised at i=N, leading to a peak at the true sample delay of the target. Moreover, using (Equation 4) and (Equation 5), we can validate
(7)$$\begin{array}{cc} \; & {\left[g\right]}_{0}\approx \underset{{\langle e\rangle }_{0}^{H}}{\underset{︸}{{\left[\bar{v_{0}};u_{0};\underline{v_{0}};\bar{v_{1}};u_{1};\underline{v_{1}}\right]}^{H}}}\left[u_{1};\bar{v_{1}};\underline{v_{0}};u_{0};\bar{v_{0}};\underline{v_{1}}\right]\approx v_{0}^{H}v_{0}+v_{1}^{H}v_{1}=2Q, \end{array}$$
where other vector multiplications are suppressed due to the independence between multipliers. The overlines and underlines are only indicators introduced to differentiate the same sub-vectors in different locations. Moreover, we note that 2Q is based on the assumption that signals carried by sub-carries have the unit amplitude. Similarly, we can obtain
(8)$$\begin{array}{cc} \; & {\left[g\right]}_{2N}\approx \underset{{\langle e\rangle }_{2N}^{H}}{\underset{︸}{{\left[\underset{N-Q}{\underset{︸}{\cdots }};\bar{v_{1}};u_{1};v_{1};\bar{v_{0}};\underset{Q}{\underset{︸}{\cdots }}\right]}^{H}}}\left[u_{1};\bar{v_{1}};\underline{v_{0}};u_{0};\bar{v_{0}};\underline{v_{1}}\right]\approx v_{0}^{H}v_{0}+v_{1}^{H}v_{1}=2Q, \end{array}$$
where the dots are entry fillers with their numbers given underneath. Using (Equation 4) and (Equation 5), we can further confirm that 2Q is the second maximum value that ${\left[g\right]}_{i}$ can take. These two CCC results are actually the false targets, as one can easily check that they are the strongest maximum CCC results after the peak ${\left[g\right]}_{N}$.

From (Equation 7), we see that the nontrivial value of ${\left[g\right]}_{0}$ is due to the *Q* samples at the tail of each essential OFDM symbol. These samples are copied to the beginning of an OFDM symbol, acting as CP. From (Equation 8), we see that the nontrivial value of ${\left[g\right]}_{2N}$ is caused by actual CPs. These observations, though based on a special case, can be generalised.

### 4.2. Analysing False Targets and Their General Features

In light of (Equation 4), we propose to decompose e as follows,
(9)$$\begin{array}{cc} \; & e=w+{\langle w\rangle }_{N}+r,\;s.t.\;w=\left[v_{0};0_{N\times 1};\cdots ;v_{\left\lfloor \frac{\tilde{N}}{N+Q}\right\rfloor -1};0_{N\times 1};0_{{\left(\tilde{N}\right)}_{N+Q}\times 1}\right], \end{array}$$
where all the *v*-vectors have the dimension of *Q*, $\tilde{N}$ is the length of each segmentation at the sensing receiver (see Algorithm 1), and ${\left(\tilde{N}\right)}_{N+Q}$ denotes the remainder of $\tilde{N}$ divided by (N+Q). Note that the first integer multiples of (N+Q) samples of w have a fixed pattern: only the first *Q* samples of each OFDM symbol (with CP) are kept.

The following properties of the generalised e will be useful for further analysis. Note that Lemma 1 can be proved using the definition of e given in (Equation 4). Lemma 2 can be confirmed based on the decomposition defined in (Equation 9).

**Lemma** **1.**
*The decomposition of e given in (Equation 9) is always valid, provided $\tilde{N}\geq N+Q$.*


**Lemma** **2.**
*The three components w, ${\langle w\rangle }_{N}$, and r have mutually exclusive positions for nonzero entries.*


Based on (Equation 9), the vector f, as obtained in Step 3 of Algorithm 1, can be written as
(10)$$\begin{array}{c} f\approx {\langle e\rangle }_{L}={\langle w\rangle }_{L}+{\langle w\rangle }_{N+L}+{\langle r\rangle }_{L}, \end{array}$$
where *L* denotes the target delay and μ=0 is continued to be assumed. As seen in Figure 1, false targets only appear in the range domain, so the Doppler information is irrelevant in the following analysis. Moreover, for illustration convenience, we continue assuming that α=1 and unit-amplitude signals are carried by sub-carriers. Furthermore, the approximation in (Equation 10) is made because the interference term from adding the virtual CP has negligible impacts on CCC. This can be inferred from (Equation 5) and (Equation 6).

As illustrated in (Equation 6), the range cut of the RDM (note that RDM is a two-dimensional matrix. The row dimension spans the range domain and the column dimension spans the Doppler domain. When we take one row out of an RDM, we obtain a so-called range cut or profile [39], as if we cut a slice of the RDM at a specific range bin. Similarly, when we take one column, we obtain a so-called Doppler cut or profile. It depicts the sensing results over range bins at a certain Doppler bin) obtained by Algorithm 1 is the CCC between e and f. Thus, the *i*-th sample of the range cut can be written as
(11)$$\begin{array}{c} {\left[g\right]}_{i}\approx {\left({\langle e\rangle }_{i}\right)}^{H}{\langle e\rangle }_{L}, \end{array}$$
where the result in (Equation 10) has been used. As expected, ${\left[g\right]}_{i}$ is maximised at i=L, where the signals involved in the CCC are fully overlapped. Moreover, using (Equation 9) and (Equation 10), we can validate that
(12)$$\begin{array}{cc} \; & {\left[g\right]}_{L+N}={\langle e\rangle }_{L+N}^{H}f\\ \;\; & =\begin{array}{c} {\langle w\rangle }_{L+N}^{H}{\langle w\rangle }_{L}+{\langle w\rangle }_{L+N}^{H}{\langle w\rangle }_{N+L}+{\langle w\rangle }_{L+N}^{H}{\langle r\rangle }_{L}+{\langle w\rangle }_{L+2N}^{H}{\langle w\rangle }_{L}+{\langle w\rangle }_{L+2N}^{H}{\langle w\rangle }_{N+L}+{\langle w\rangle }_{L+2N}^{H}{\langle r\rangle }_{L}\\ +{\langle r\rangle }_{L+N}^{H}{\langle w\rangle }_{L}+{\langle r\rangle }_{L+N}^{H}{\langle w\rangle }_{N+L}+{\langle r\rangle }_{L+N}^{H}{\langle r\rangle }_{L} \end{array}\\ \; & ={\langle w\rangle }_{L+N}^{H}{\langle w\rangle }_{N+L}+\epsilon_{L+N}=\left\lfloor \frac{\tilde{N}}{N+Q}\right\rfloor Q+\epsilon_{L+N}, \end{array}$$
where the other terms are zero based on Lemma 2. Note that $\epsilon_{L+N}$ is a residual term, as given by
$$\epsilon_{L+N}={\langle w\rangle }_{L+2N}^{H}{\langle w\rangle }_{L}+{\langle w\rangle }_{L+2N}^{H}{\langle r\rangle }_{L}+{\langle r\rangle }_{L+N}^{H}{\langle r\rangle }_{L}.$$

Similar to ${\left[g\right]}_{L+N}$, we can obtain
(13)$$\begin{array}{c} {\left[g\right]}_{L-N}={\langle w\rangle }_{L}^{H}{\langle w\rangle }_{L}+\epsilon_{L-N}=\left\lfloor \frac{\tilde{N}}{N+Q}\right\rfloor Q+\epsilon_{L-N}, \end{array}$$
where $\epsilon_{L-N}$ is a residual term similar to $\epsilon_{L+N}$. Although $\epsilon_{L-N}$ and $\epsilon_{L+N}$ cannot be analytically expressed, we can safely assume that their values are small, due to their noise-like and (partially) independent entries.

Similar to (Equation 12) and (Equation 13), one can easily calculate ${\left[g\right]}_{i}$ at other is. Given the noise-like signals in e (see Remark 1) and Lemmas 1 and 2, we can further validate that ${\left[g\right]}_{L\pm N}$ have the largest amplitudes, only next to the global peak ${\left[g\right]}_{L}$, among all CCC results. We summarise the above analysis formally into the following proposition.

**Proposition** **1.**
*Given $\tilde{N}\geq \left(N+Q\right)$, USF can present two strong false targets accompanying a true target in the range domain. They locate at $\left[{\left(L+N-1\right)}_{\tilde{N}}+1\right]$-th and the $\left[{\left(L-N-1\right)}_{\tilde{N}}+1\right]$-th range bins, where L is the true target delay, and “+1” and “−1” are because the starting index is one. The magnitudes of the two false targets are approximately identical. The value approximates $\left\lfloor \frac{\tilde{N}}{N+Q}\right\rfloor \frac{Q}{\tilde{N}}$, when normalised by the peak magnitude of the true target.*


**Proof.** The analysis in Section 4.2 proves the results in the proposition for the case with α=1 and unit-amplitude communication signals over sub-carriers. Here, we mainly show that the proposition holds without the two assumptions.From (Equation 12) and (Equation 13), we see that the locations of the false targets mainly depend on the locations of nonzero entries in w. Thus, we only need to confirm the amplitude relation between the false targets and the true one.There can be two cases when the communication signals over OFDM sub-carriers have nonconstant amplitudes. First, constellations such as quadrature amplitude modulation (QAM) are used. Second, the single-carrier OFDM is performed. It means that the signals over sub-carriers are from the discrete Fourier transform of signals drawn from some constellation. In either case, the mean of the communication signals over sub-carriers is zero. Thus, the squared sum of the signals, normalised by the number of samples, approximates the variance of some statistic. This further leads to ${\left[g\right]}_{L}\approx \alpha \tilde{N}\sigma^{2}$, where $\sigma^{2}$ denotes the variance of the signals over sub-carriers. Similarly, based on (Equation 12) and (Equation 13), we have ${\left[g\right]}_{L\pm N}\approx \alpha \left\lfloor \frac{\tilde{N}}{N+Q}\right\rfloor Q\sigma^{2}$, where the ϵ-terms are suppressed here. Therefore, the ratio between ${\left[g\right]}_{L\pm N}$ and ${\left[g\right]}_{L}$ is $\left\lfloor \frac{\tilde{N}}{N+Q}\right\rfloor \frac{Q}{\tilde{N}}$, as stated in Proposition 1.    □

## 5. Designing Methods to Remove False Targets

We proceed to design new methods for removing false targets. Two solutions are provided in this section. Their block diagrams are illustrated in Figure 3. The first solution, as developed in Section 5.1, seeks to remove the false targets completely by preventing them from being generated. This is mainly achieved by the first two steps shown in Figure 3a. The second solution allows the false targets to happen and then employs their features unveiled in Proposition 1 to remove them. The block diagram of the second solution is given in Figure 3b, with details of the solution illustrated in Section 5.2.

### 5.1. Removing False Targets by Nullifying CP Signals

From Section 4, we see that the false targets are caused by w and ${\langle w\rangle }_{N}$ in e. Recall that e is the first column of $\overset{˘}{X}$, as obtained in Step 2 of Algorithm 1. Similar to e, other columns of $\overset{˘}{X}$ can also be decomposed, as performed in (Equation 9), and there will be w-like vectors leading to false targets. From (Equation 9), we see that the nonzero entries in w are actually CPs of OFDM signals. Therefore, a simple way of preventing false targets from being generated is to remove CPs, such as w, and the signals used for generating CPs, such as ${\left(w\right)}_{N}$, in $\overset{˘}{X}$. The feasibility of modifying $\overset{˘}{X}$ is ensured by the fact that $\overset{˘}{X}$ is constructed at the sensing receiver using the copy of communication-transmitted signals in the time domain. In other words, whatever changes made to $\overset{˘}{X}$ will not affect communications.

Due to the reshaping performed in Step 2 of Algorithm 1, CPs will fall in different locations over the columns of $\overset{˘}{X}$. However, they are traceable by referring to the indexes of the entries in x that form a column of $\overset{˘}{X}$. Specifically, according to Definition 1, we know that the *m*-th column of $\overset{˘}{X}$ consists of entries in x with indexes given by
(14)$$\begin{array}{c} I_{m}=\left\{\left(m-1\right)\left(\tilde{N}-\bar{Q}\right)+\left[1,2,\cdots ,\tilde{N}\right]\right\},\;m=1,\cdots ,\left\lfloor \frac{I-\bar{Q}}{\tilde{N}-\bar{Q}}\right\rfloor , \end{array}$$
where I(=M(N+Q)) is the total number of samples in x. The index of original CPs in x can be given by
(15)$$\begin{array}{c} I_{\mathrm{CP}}=\left\{I_{\mathrm{CP}}^{m},\;m=0,1,\cdots ,M-1\right\}, \end{array}$$
where $I_{\mathrm{CP}}^{m}$ is the set of CP indexes in the *m*-th OFDM symbol. It can be given by
(16)$$\begin{array}{c} I_{\mathrm{CP}}^{m}=\left\{\left(1,2,\cdots ,Q,N+1,\cdots ,N+Q\right)+m\left(N+Q\right)\right\}. \end{array}$$
Note that not only CPs but also the samples at the tail of each OFDM symbol that are used as CPs contribute to the presence of false targets.

To nullify signals in $\overset{˘}{X}$ that are related to CPs in the original OFDM signals, we introduce an indicating matrix W of the same size as $\overset{˘}{X}$ and initiate it with all ones. Then, for $m=0,1,\cdots ,\left\lfloor \frac{I-\bar{Q}}{\tilde{N}-\bar{Q}}\right\rfloor$, we identify the entries in $I_{m}$ that are also in $I_{\mathrm{CP}}$. At the indexes of those entries, we set the entries of W in the *m*-th column as zeros. After enumerating all values of *m*, we set $\overset{˘}{X}$ in Algorithm 1 by $\overset{˘}{X}⊙W$ before performing Steps 3–7 to prevent false targets from being generated. The above illustration is summarised in Algorithm 3.
**Algorithm** **3** Modified USF (MUSF).1.        Run Steps 1–2 of Algorithm 1, obtaining Z and $\overset{˘}{X}$;2.        Set $W=1_{\tilde{N}\times \tilde{M}}$, with the same dimension as $\overset{˘}{X}$;3.        For $m=1,\cdots ,\left\lfloor \frac{I-\bar{Q}}{\tilde{N}-\bar{Q}}\right\rfloor$            (a)        Find entries in $I_{m}$, as given in (Equation 14), that also exist in $I_{\mathrm{CP}}$, as given in (Equation 15), and collect their indexes in I;            (b)        Set ${\left[W\right]}_{I,m}=0$, where ${\left[W\right]}_{I,m}$ denotes the *m*-th column of W with rows indexed by entries in I;4.        Set $\overset{˘}{X}=\overset{˘}{X}⊙W$;5.        Run Steps 3–7 of Algorithm 1, yielding an RDM without false targets;


Nullifying CPs prevents the false targets from being generated. However, it will also reduce the processing gain on the true target. This can be seen from (Equation 6). After nullifying CPs, the first e on the right-hand side of the approximation in (Equation 6) becomes $e⊙{\left[W\right]}_{:,1}$, where ${\left[W\right]}_{:,1}$ denotes the first column of W obtained after performing Step 3 of Algorithm 3. Since ${\left[W\right]}_{:,1}$ consists of either ones or zeros, ${\left[g\right]}_{i}$ in (Equation 6) is still maximised at i=L, yet with a smaller peak amplitude. This weakened peak can reduce the detecting probability of the true target. Next, we develop another method for handling false targets.

### 5.2. Sequential Target Detection and Removal

Unlike the previous method preventing false targets from appearing, this method handles the false targets using the properties unveiled in Proposition 1. In particular, as stated therein, false targets actually have known locations relative to that of the true target. Moreover, the relation between the strengths of true and a false target is also known. Utilising these features, we develop a sequential target detection and removal method, based on the RDM $\overset{˘}{Z}$ obtained in Step 7 of Algorithm 1.

We start with searching for the strongest peak in $\overset{˘}{Z}$:(17)$$\begin{array}{cc} \left(n_{t},m_{t}\right):\; & \arg\max_{n,m}\left|{\left[\overset{˘}{Z}\right]}_{n,m}\right|,\;n=1,\cdots ,\tilde{N},\;m=1,\cdots ,\tilde{M}, \end{array}$$
where $\left(n_{t},m_{t}\right)$ is the two-dimensional index in the range-Doppler-domain, and the subscript *t* denotes the target index. Treating the peak at $\left(n_{t},m_{t}\right)$ as a true target, we can deduce the features of the accompanying false targets. According to Proposition 1, the locations of false targets are
(18)$$\begin{array}{c} \left(n_{t}^{\pm },m_{t}\right)=\left({\left(n_{t}\pm N-1\right)}_{\tilde{N}}+1,m_{t}\right), \end{array}$$
where ‘−1’ and ‘1’ are because the index starts from one here. Moreover, the amplitudes at the local peaks of the two false targets are the scaled version of the peak amplitude at the true target, specifically,
(19)$$\begin{array}{c} P_{\left(n_{t}^{\pm },m_{t}\right)}=P_{t}k,\;\;s.t.\;k=\left\lfloor \frac{\tilde{N}}{N+Q}\right\rfloor \frac{Q}{\tilde{N}},\;P_{t}={\left[\overset{˘}{Z}\right]}_{n_{t},m_{t}}, \end{array}$$
where $P_{t}$ is the amplitude of the true target. Note that $\left\lfloor \frac{\tilde{N}}{N+Q}\right\rfloor \frac{Q}{\tilde{N}}$ is the ratio depicting the amplitude relation between true and false targets, as derived in Proposition 1. If the target range and Doppler frequencies are integer multiples of their respective resolutions, we can reconstruct an RDM related to the strongest target and remove it from the $\overset{˘}{Z}$. However, it is likely that the target range and Doppler frequencies are noninteger multiples of their respective resolutions. To account for this case, it can be better to also include the neighbour bins surrounding the peaks of true and false targets when reconstructing the RDM for target removal.

Let $N_{r}$$\left(N_{d}\right)$ denote the number of the single-side range (Doppler) bins to be included for target reconstruction. Then, the set of indexes for the range-Doppler bins, where the true target is located, can be given by
(20)$$\begin{array}{c} I_{t}=\left\{\left(n_{t}+i,m_{t}+j\right),\;i=-N_{r},\cdots ,0,\cdots ,N_{r},\;j=-N_{d},\cdots ,0,\cdots ,N_{d}\right\}. \end{array}$$

The set of indexes for the range-Doppler bins, where the false targets are located, can be given by
(21)$$\begin{array}{c} I_{t}^{\pm }=\left\{\left({\left(n_{t}\pm N-1\right)}_{\tilde{N}}+1+i,m_{t}+j\right),\;i=-N_{r},\cdots ,0,N_{r},\;j=-N_{d},\cdots ,0,N_{d}\right\}. \end{array}$$

The RDM of the target can be reconstructed as
(22)$$\begin{array}{cc} \; & {\left[Z_{t}\right]}_{n,m}=\left\{\begin{array}{cc} {\left[\overset{˘}{Z}\right]}_{n,m}, & \mathrm{if}\;\left(n,m\right)\in \left\{I_{t}\right\}\\ {\left[\overset{˘}{Z}\right]}_{n^{\prime },m^{\prime }}k, & \mathrm{if}\;\left(n,m\right)\in \left\{I_{t}^{\pm }\right\}\\ 0, & \mathrm{otherwise} \end{array}\right., \end{array}$$
where $\left(n,m\right)=\left({\left(n^{\prime }\pm N-1\right)}_{\tilde{N}}+1,m^{\prime }\right)$ for the second case, and k is given in (Equation 19). Note that the amplitudes for the false targets are estimated based on the their properties derived in Proposition 1. To suppress the impact of the false targets associated with the strongest target on the detection of other targets, we can update $\overset{˘}{Z}$ as
(23)$$\begin{array}{c} \overset{˘}{Z}=\overset{˘}{Z}-Z_{t}. \end{array}$$
Then, we detect the next strongest target, reconstruct it, and remove it, as illustrated above. The sequential target detection and removal method is summarised in Algorithm 4.
**Algorithm** **4** Sequential target detection and removal.Input: the number of targets *T*, $N_{r}$, $N_{d}$1.        Run Algorithm 1, obtaining $\overset{˘}{Z}$;2.        For t=1,⋯,T           (a)        Get $\left(n_{t},m_{t}\right)$ using (Equation 17);           (b)        Construct $I_{t}$ and $I_{t}^{\pm }$ based on (Equation 20) and (Equation 21), respectively;           (c)        Construct the single-target RDM $Z_{t}$ as per (Equation 22);           (d)        Update $\overset{˘}{Z}=\overset{˘}{Z}-Z_{t}$;

In Algorithm 4, we assume that the number of overall targets is known, for illustration convenience. In practice, we can perform a threshold detection to determine whether a maximum peak in the RDM is a target or not. Numerous methods are available for determining the threshold, such as the Neyman–Pearson criterion and the constant false alarm rate (CFAR) detector, etc. One may refer to classical radar texts, e.g., [39], for more details on this topic. In simulations, we will employ the cell averaging CFAR (CA-CFAR) [39] [ch.16.5] to fulfil the task.

Next, we remark on the computational complexity (CC) of the two algorithms developed in this section. For Algorithm 3, the steps for nullifying CPs, namely, Steps 2–4, have negligible CC compared with other sensing steps. This is because the indexes of CPs are easily traceable and nullifying them does not involve any computations. Therefore, Algorithm 3 has a CC in the same order as Algorithm 1, which is $O\left(\tilde{N}\tilde{M}\log\left(\tilde{N}\tilde{M}\right)\right)$ [32].

For Algorithm 4, we notice that Step 1 runs Algorithm 1 and hence has a complexity of $O\left(\tilde{N}\tilde{M}\log\left(\tilde{N}\tilde{M}\right)\right)$. Moreover, Step 2a, which searches the peak of the RDM $\overset{˘}{Z}$, contributes most to the CC of the overall Step 2. Thus, given *T* iterations, the overall CC of Step 2 is in the order of $O\left(T\log\left(\tilde{N}\tilde{M}\right)\right)$. We note that *T* is theoretically equal to the number of targets and hence is much smaller than $\tilde{N}\tilde{M}$. Thus, we can conclude that Algorithm 4 also has a CC of $O\left(\tilde{N}\tilde{M}\log\left(\tilde{N}\tilde{M}\right)\right)$.

## 6. Simulation Results

In this section, simulation results are provided to validate the effectiveness of the proposed method. First, we continue the scenario given in Table 1 to illustrate how the proposed methods impact RDMs.

Figure 4a plots the RDM of MUSF given in Algorithm 3, where parameters given in Table 1 are reused here. We see that, substantially different from USF-RDM in Figure 1a, the false targets are removed in MUSF-RDM. Figure 4b plots the RDM after running Algorithm 4. Since the algorithm removes the strongest target sequentially, the residual RDM would be target-free, if the number of targets is known. This is validated by Figure 4b. We see that the residual RDM is noise-like without obvious target peaks. Figure 4b also confirms the features of false targets derived in Proposition 1, as false targets are reconstructed based on those features.

Figure 5 further compares the range and Doppler cuts of different algorithms. From Figure 5a, we see that the proposed Algorithms 3 and 4 can both remove the false targets. From Figure 5b, we see that the two algorithms have negligible impacts on the Doppler spectrum. This is consistent with our analysis in Section 4. Moreover, we can see from Figure 5a that Algorithm 3 has slightly larger sidelobes than the other two algorithms overall. This is actually caused by the reduced peak magnitude of Algorithm 3. Since the peak magnitude is normalised to one in the figure, the sidelobe level is raised as a result.

Next, we increase the number of targets to four to validate the proposed methods. With other parameters in Table 1 fixed, the target delays of the four targets are set as 312,262,375, and 382. Moreover, their respective Doppler frequencies are 195.06 Hz, −650.76 Hz, −476.77 Hz, and 709.80 Hz. These are randomly generated without any special implications.

Figure 6a plots the RDM obtained by COS, as given in Algorithm 2. We see that the classical COS, as fully complied with the underlying communication system, fails to generate a normal RDM with peaks at the target locations. Figure 6b plots the RDM of USF, as given in Algorithm 1. Other than the four true targets highlighted, we also see eight false targets (two associated with each true target). Figure 6c presents the RDM of the modified USF given in Algorithm 3. We see that the false targets are effectively removed.

Figure 7 plots the updated RDMs in the first three iterations of Algorithm 4. Note that with four targets set, the algorithm has four iterations in total. From the three RDMs, we can see that each iteration is able to remove one true target along with its associated false targets. This again validates the correctness of our analysis of the features of the false targets. Moreover, this manifests the effectiveness of Algorithm 4 in constructing the RDM of a target and removing it from the overall RDM.

Next, we employ the common metrics to further highlight the impacts of false targets and the proposed methods on sensing. In particular, we perform CA-CFAR based on the RDM obtained under different methods, and observe the detecting probability as well as the false alarm rate (when there is no target but the radar detection reports a target, a false alarm happens. The average number of false alarms per unit time is defined as the false alarm rate [39]) against the signal-to-noise ratio (SNR). We also consider single- and multitarget cases. In both cases, target delay, velocities, and amplitudes are randomly generated over $10^{4}$ independent trials. Moreover, the target delay is in the range of [N,1.5N], where N=256 is the original OFDM length. The target speed is in the range of [−120,120] m/s. Further, the target amplitude conforms to a complex Gaussian distribution with the mean of one and the variance of 0.01.

A key parameter of CA-CFAR is the expected false alarm rate. It is used in computing the detecting threshold. Here, we set the expected false alarm rates for Algorithms 1–4 as $10^{-4}$, $10^{-5}$, $10^{-5}$, and $10^{-6}$, respectively. In general, the smaller the value, the greater the detecting threshold would be. As a result, the detecting probability would be reduced.

Figure 8 plots the detecting performance achieved based on the RDMs obtained using different methods. From Figure 8a, we see that Algorithm 4 achieves the maximum detecting probability. This is mainly because we simply detect the maximum peaks as targets without applying threshold detection, as do the other three algorithms. However, identifying the maximum peak as target is enabled by our proposed Algorithm 4. It effectively remove the impacts of stronger targets and their associated false targets on weaker ones.

From Figure 8a, we also see that COS basically fails to detect targets in the whole SNR region. Moreover, we see that Algorithm 1 always achieves higher detecting probabilities than Algorithm 3. This reveals a disadvantage of Algorithm 3. It indeed prevents the false targets from being generated by nullifying CPs, but it also reduces the processing gains on the true targets due to the signal nullification.

From Figure 8b, we see that Algorithms 3 and 4, as developed in this work, can reduce the false alarm rate, compared with Algorithm 1, which suffers from false targets. This validates that the proposed algorithms can effectively counteract the impacts of false targets. Moreover, Algorithms 1 and 3 have higher false alarm rate than Algorithm 2, although the former two algorithms actually use a smaller theoretical false alarm rate to calculate the detecting threshold than the third one. This is a consequence of not only false targets but also the virtual CPs introduced in the sensing framework of Algorithm 1. As analysed in [32], the background noise level can be slightly raised by Algorithm 1 compared with the classical COS given in Algorithm 2. Moreover, there can be a fixed noise floor even when the receiver noise is negligibly weak. All these consequences contribute to the slightly increased false alarm rate of Algorithm 1 and its variants Algorithms 3 and 4.

In parallel, Figure 8 and Figure 9 presentthe detecting performance achieved based on different RDMs. From Figure 9a, we see that Algorithm 4 substantially outperforms the other three algorithms. This is because Algorithm 4 has the mechanism to reduce interference of strong targets on weaker ones. Again, the feasibility of doing so is enabled by our discovery on false target features in this work. From Figure 9b, we further see that the proposed Algorithms 3 and 4 can effectively reduce the false alarm rate.

Table 2 compares the four algorithms simulated above by highlighting their key features/results. We see that the proposed Algorithms 3 and 4 are able to sense longer distances than the benchmark Algorithm 2 [21]. We also see that the proposed Algorithms 3 and 4 can remove false targets with lower false alarm rates achieved, as compared with the benchmark Algorithm 1 [32]. Moreover, we see that the proposed Algorithm 4 not only reduces false alarm rate but also maintains a high detecting probability, while Algorithm 3 reduces the false alarm rate at the cost of decreasing its detecting probability. In addition, comparing the first and last rows in the table, we see that Algorithm 4 is able to reduce the false alarm rate by over 50% of that achieved by Algorithm 1.

## 7. Conclusions

In this paper, we reveal the reasons for false targets in USF and develop a solution to the issue. Specifically, we illustrate that false targets are generated by periodically appearing CP signals. We unveil that false targets have the same Doppler bin but different range bins compared with the true target. Moreover, the differences between the range bins of false and true targets are derived. Other features, such as strength, of false targets are also provided. In addition, we propose to remove false targets by nullifying CP signals of a coefficient matrix used for processing echo signals. We also propose a second method performing sequential target detection and removal to counteract the impact of false targets. Extensive simulations are performed, validating the effectiveness of the proposed methods. The proposed algorithm can reduce the false alarm rate by more than 50% compared with the previous work [32]. Meanwhile, the detecting probability can be maintained at one for a large SNR region.

## Figures and Tables

**Figure 1 sensors-22-09015-f001:**
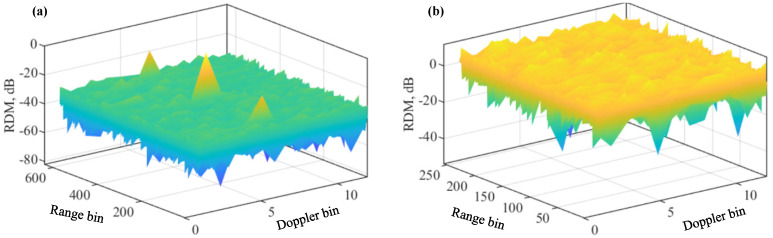
Illustrating the RDMs of UFS (**a**) and COS (**b**). Please note that the symbol “-” before all numbers, as automatically generated by MATLAB during plotting the figure, denotes the negative sign.

**Figure 2 sensors-22-09015-f002:**
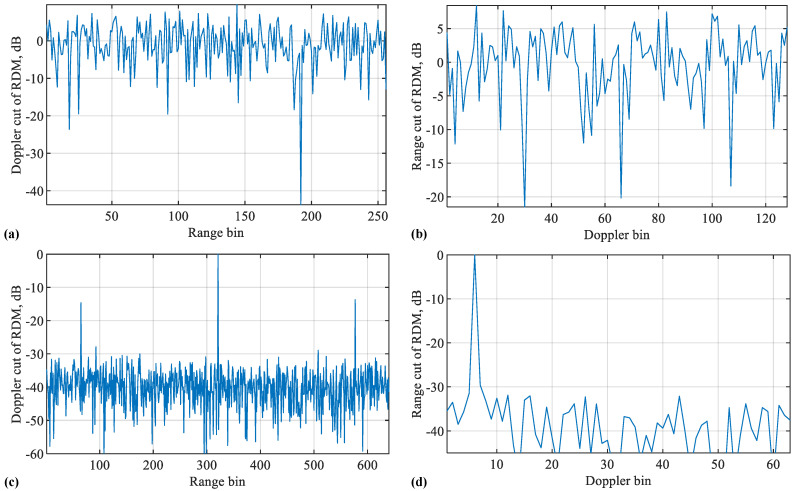
Doppler (**a**) and range (**b**) cuts of the RDM obtained by COS, as plotted in Figure 1b. Doppler (**c**) and range (**d**) cuts of the RDM obtained by USF, as plotted in Figure 1a. Please note that the symbol “-” before all numbers, as automatically generated by MATLAB during plotting the figure, denotes the negative sign.

**Figure 3 sensors-22-09015-f003:**
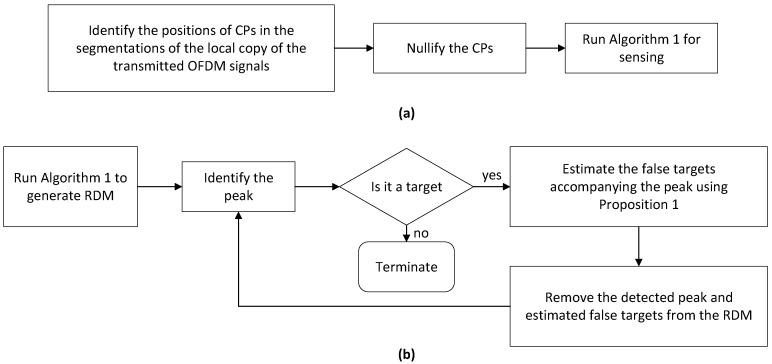
(**a**) The block diagram for the solution developed in Section 5.1; (**b**) the block diagram for the method designed in Section 5.2.

**Figure 4 sensors-22-09015-f004:**
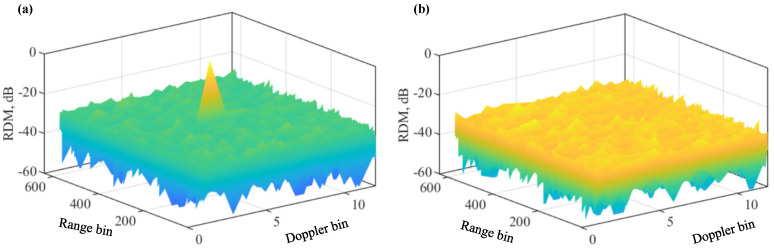
RDM of Algorithm 3 in (**a**); RDM of Algorithm 4 in (**b**). A single target is set here. Please note that the symbol “-” before all numbers, as automatically generated by MATLAB during plotting the figure, denotes the negative sign.

**Figure 5 sensors-22-09015-f005:**
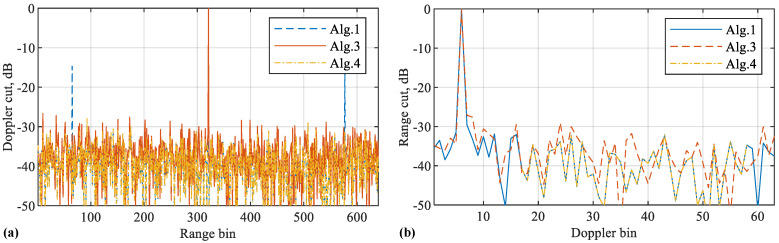
(**a**) Doppler cuts of the RDMs shown in Figure 4; (**b**) range cuts of the same RDMs. Note that Algorithm 1 as developed in [32], is simulated as a benchmark. It suffers from the false target issue. In contrast, the new algorithms, Algorithms 3 and 4, are able to remove the false targets. Please note that the symbol “-” before all numbers, as automatically generated by MATLAB during plotting the figure, denotes the negative sign.

**Figure 6 sensors-22-09015-f006:**
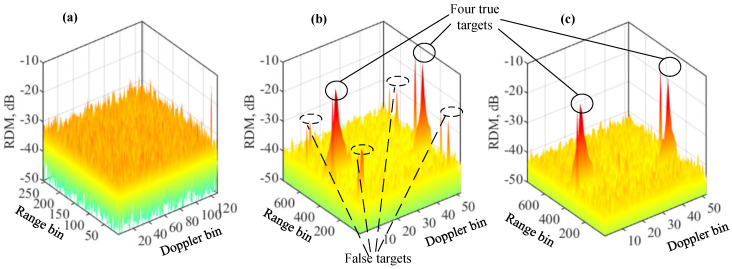
RDM of COS in (**a**); RDM of Algorithm 1 in (**b**); RDM of Algorithm 3 in (**c**). Four targets are set, as highlighted in the figures. Please note that the symbol “-” before all numbers, as automatically generated by MATLAB during plotting the figure, denotes the negative sign.

**Figure 7 sensors-22-09015-f007:**
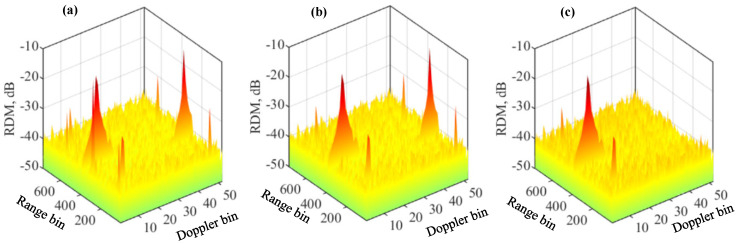
RDMs obtained by Algorithm 4 with four targets simulated. The RDMs from the first three iterations in the algorithms are given in (**a**–**c**), respectively. Please note that the symbol “-” before all numbers, as automatically generated by MATLAB during plotting the figure, denotes the negative sign.

**Figure 8 sensors-22-09015-f008:**
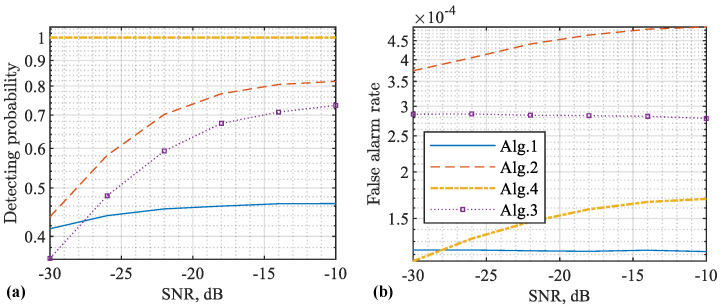
Detecting performance versus SNR (with respect to y), where a single target is set with randomly generated parameters over $10^{4}$ independent trials. The detecting probability is given in (**a**) and the false alarm rate is given in (**b**). Note that Algorithms 1 and 2, as developed in [21,32], respectively, are simulated as benchmark methods. Additionally note that Algorithm 2 [21] cannot properly sense a target with the echo delay over the CP duration, and hence its detecting probability is much lower than our algorithms. Please note that the symbol “-” before all numbers, as automatically generated by MATLAB during plotting the figure, denotes the negative sign.

**Figure 9 sensors-22-09015-f009:**
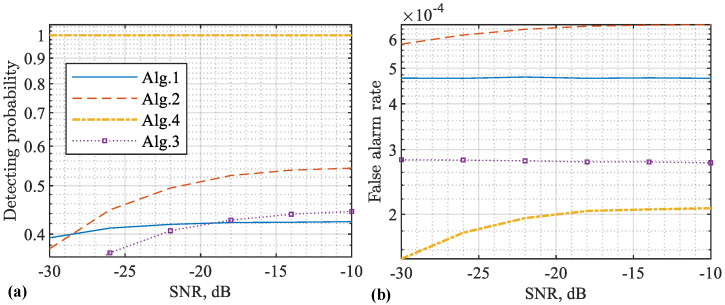
Detecting performance versus SNR (with respect to y), where four targets are set with randomly generated parameters over $10^{4}$ independent trials. The detecting probability is given in (**a**) and the false alarm rate is given in (**b**). Please note that the symbol “-” before all numbers, as automatically generated by MATLAB during plotting the figure, denotes the negative sign.

**Table 1 sensors-22-09015-t001:** Simulation parameters.

Variable	Description	Value
$f_{c}$	Carrier frequency	2.4 GHz
*B*	Bandwidth	3.84 MHz
*M*	No. of OFDM symbols	128
*Q*	CP length	64
*N*	No. of sub-carriers	256
*L*	Target sample delay	320(=N+Q)
μ	Doppler frequency	480 Hz
|α|	Signal amplitude in y; see (Equation 3)	0 dB
–	Noise power in y	30 dB
$\left[\tilde{Q},\tilde{N},\bar{Q}\right]$	Parameters of USF	$\left[L,2\tilde{Q},0\right]$

**Table 2 sensors-22-09015-t002:** Comparisons among different algorithms.

Algorithm	Long-Range Sensing (Figure 6)	False Targets (Figure 5)	Detection Probability (−10 dB, Figure 9)	False Alarm Rate (−10 dB, Figure 9)
1	yes	yes	0.54	$4.8\times 10^{-4}$
2	no	no	0.42	$6.3\times 10^{-4}$
3	yes	no	0.44	$2.8\times 10^{-4}$
4	yes	no	1	$2.05\times 10^{-4}$

## Data Availability

Not applicable.

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
