# Peer review of "Removing False Targets for Cyclic Prefixed OFDM Sensing with Extended Rangingâ€"

_sensors, 2022, doi:10.3390/s22229015_

Round 1
Reviewer 1 Report
Thank you for you interesting work.
I have some (MINOR) comments that must be considered in the revised manuscript.
-----------------------------------
1) The note (1) which declares that this work is an extension for a previous work by same authors can be modified as follows.
In the objective (at the end of "Introduction", authors can write:
In a previous work we did so and so.... In the new work, we make so and so..... differentiate between the 2 works.
2) Both Abstract and Conclusion miss some (main) numerical results.
3) Explain what do you mean with (cut of). This word appears once and the text and many times in the figures (Ex: Doppler cut of.... Fig. 2+4).
4) The procedure is better to be summarized in a flow chart (or a block diagram) before dealing with details.
5) Many equations need a reference (Ex: Eq. 1 + Eq. 3).
6) Explain the word (bin) before using it. It is used in "range" and in "Doppler".
7) How can you judge the correctness of your results. I cannot see any comparison with a previously published work.
8) Figure 4 illustrates results using different algorithms. Can you please compare (in a summarizing table) the improvement (or enhancement) achieved with each algorithm.
9) I do NOT like figures in the "Conclusion". I see Fig. 8 belongs to the results and discussion (NOT conclusion). Please transfer before Conclusion.
10) Define the (False alarm rate) before use.
Author Response
We thank this reviewer for your time and effort in reviewing our paper. Your valuable comments have all been addressed and reflected in the revised paper. Please refer to the response letter and the revised paper for more details. Major changes are highlighted in blue colour for your convenience.

Reviewer 2 Report
The paper investigates and attempts to solve the problem of false targets in USF. In the article, the Authors show that false targets are caused by periodic CP cyclic prefixes in the CP-OFDM waveform, and that false targets have the same Doppler bin but different range bins compared to the true target. They determine the relationship between the location of false and true targets and other characteristics, such as the strength of false targets. The paper proposes an effective solution to remove false targets by nullifying the CP signals of the coefficient matrix used to process echo signals. The second method presented performs sequential target detection and removal to counteract the influence of false targets. The simulations performed confirm the validity of the realized analysis, as well as the effectiveness of the proposed solution.
From the point of view of the entire work, it is advisable to expand the introduction in order to show a more extensive literature review. Also, it is worth expanding this review because many of the cited literature items are authored by the Authors of this work. The paper presents an extensive mathematical apparatus and is clearly scientific in nature. I would still recommend revising the illustrations for readability of the presented content and expanding the summary section.
Author Response
We thank this reviewer for your time and effort in reviewing our paper. We have expanded the introduction per your suggestion. Please see the response letter and the revised paper for more details. Major changes are highlighted in blue colour for your convenience.

Reviewer 3 Report
This paper provides an algorithm for Removing False Targets for Cyclic Prefixed OFDM Sensing with Extended Ranging. The paper needs to be rewritten in readable version. The author needs to summarize the paper in the introduction, also they need to calculate the algorithm complexity and compare their results with our schemes. Reviewers have a comment about the CP length effect on the results and accuracy. Also, what about the corruption level effect of the CP and have it can effect on the results.
The authors can refer to these refence if they see it useful:
1- Real Signal DHT-OFDM With Index Modulation for Underwater Acoustic Communication
2- Energy Harvesting for TDS-OFDM in NOMA-Based Underwater Communication Systems
3- Sea Experimental for Compressive Sensing-Based Sparse Channel Estimation of Underwater Acoustic TDS-OFDM System
4- Precoded IM-OFDM-SS for Underwater Acoustic Communication
5- Underwater TDOA acoustical location based on majorization-minimization optimization
